# Electrochemical Reduction of CO_2_: A Review of Cobalt Based Catalysts for Carbon Dioxide Conversion to Fuels

**DOI:** 10.3390/nano11082029

**Published:** 2021-08-09

**Authors:** Muhammad Usman, Muhammad Humayun, Mustapha D. Garba, Latif Ullah, Zonish Zeb, Aasif Helal, Munzir H. Suliman, Bandar Y. Alfaifi, Naseem Iqbal, Maryam Abdinejad, Asif Ali Tahir, Habib Ullah

**Affiliations:** 1Center of Research Excellence in Nanotechnology, King Fahd University of Petroleum and Minerals (KFUPM), Dhahran 31261, Saudi Arabia; aasifh@kfupm.edu.sa (A.H.); aboalzoz4ever@gmail.com (M.H.S.); bandar_alfaify@hotmail.com (B.Y.A.); 2Wuhan National Laboratory for Optoelectronics, Huazhong University of Science and Technology, Wuhan 430074, China; 2017511018@hust.edu.cn; 3Department of Chemistry, University of Glasgow, Glasgow G12 8QQ, UK; mustcapture@yahoo.com; 4College of Chemistry and Molecular Engineering, Peking University, Beijing 100871, China; latifucas@hotmail.com; 5Key Laboratory of Organic Optoelectronics & Molecular Engineering of Ministry of Education, Department of Chemistry, Tsinghua University, Beijing 100084, China; zns18@mails.tsinghua.edu.cn; 6US-Pakistan Centre for Advanced Studies in Energy (USPCAS-E), National University of Sciences and Technology (NUST), Islamabad 44000, Pakistan; naseem@uspcase.nust.edu.pk; 7Department of Physical and Environmental Sciences, University of Toronto Scarborough, 1265 Military Trail, Toronto, ON M1C 1A4, Canada; M.Abdinejad@tudelft.nl; 8Environment and Sustainability Institute, University of Exeter, Penryn, Cornwall TR10 9FE, UK; A.Tahir@exeter.ac.uk

**Keywords:** CO_2_ conversion, electrocatalysts, cobalt catalysts, MOFs, ECO_2_RR

## Abstract

Electrochemical CO_2_ reduction reaction (CO_2_RR) provides a promising approach to curbing harmful emissions contributing to global warming. However, several challenges hinder the commercialization of this technology, including high overpotentials, electrode instability, and low Faradic efficiencies of desirable products. Several materials have been developed to overcome these challenges. This mini-review discusses the recent performance of various cobalt (Co) electrocatalysts, including Co-single atom, Co-multi metals, Co-complexes, Co-based metal–organic frameworks (MOFs), Co-based covalent organic frameworks (COFs), Co-nitrides, and Co-oxides. These materials are reviewed with respect to their stability of facilitating CO_2_ conversion to valuable products, and a summary of the current literature is highlighted, along with future perspectives for the development of efficient CO_2_RR.

## 1. Introduction

The excessive combustion of fossil fuels has caused massive carbon dioxide (CO_2_) emissions, leading to rapid global environmental changes such as global warming, air pollution, desertification, acid rains, rise in sea levels, and extreme weather conditions [1]. The threats to human life and the environment due to high CO_2_ emissions are increasing day by day with growing energy demands. The kneeling curve from the Mona lova observatory shows a drastic increase in CO_2_ concentration. In May 2021, the CO_2_ concentration was 419.13 ppm which is caused due to enormous emission of CO_2_ into the atmosphere (40 billion metric tons per year). According to the Global Warming Potential (GWP) updates by the United States Environmental Protection Agency (EPA), CO_2_ has a GWP value of 1, which is lower than the value of CH_4_, N_2_O, chlorofluorocarbons (CFCs), hydrofluorocarbons (HFCs), and other greenhouse gases. However, the CO_2_ absorption energy is much higher as it has remained for thousands of years in the atmosphere compared to other greenhouse gases, reflecting the severe contribution of CO_2_ in global warming. To prevent these harmful impacts, the Paris Climate Agreement was signed by United Nations to limit the rise in temperature to 2 °C in the 21st century above the pre-industrial levels and pursuing to reduce it to 1.5 °C. However, the predictions by the US Energy Information Administration state that the use of fossil fuels is expected to increase until 2040, which will cause the global temperature to rise more than 2 °C. CO_2_ gas emission should be equal to the amount of CO_2_ consumed. Therefore, alternatives to reduce global CO_2_ emissions and CO_2_ conversions to less harmful products are highly needed to cope with global warming issues in time [2,3].

Several carbon capture and utilization methods are implemented to mitigate CO_2_ concentration in the atmosphere and overcome its environmental challenges [4,5,6,7,8]. The main strategies to reduce CO_2_ emissions deal with the circular carbon economy (CCE), a holistic approach that consists of Reduce, Reuse, Recycle and Remove (4Rs) of CO_2_. The reuse of CO_2_ is categorized to search for low carbon energy alternatives such as wind, solar and hydro energy for replacing fossil fuels. Another approach is geological sequestration, a promising strategy to provide a low carbon energy future [9,10,11]. Still, there is uncertainty about stored CO_2_ for a long time, and it might have leakage issues. Another approach is recycling, and utilizing CO_2_ into other useful chemicals is the most attractive strategy to reduce CO_2_ emissions [12,13,14]. Catalysis plays a vital role in our daily life. Various types of catalysts have been reported for the conversion of waste into useful products, including zeolites [15,16,17,18,19,20,21,22,23,24,25,26,27,28,29], metal and metal oxides [30,31,32,33,34,35,36,37,38,39,40,41,42], nitrides [43,44,45,46,47,48,49,50,51], carbon-based catalysts [52,53,54,55,56,57,58,59,60,61,62,63], metal complexes [64,65,66,67,68,69,70,71,72] or highly porous metal–organic frameworks (MOFs) and covalent organic frameworks (COFs) [5,73,74,75,76,77,78,79,80]. The synthesis of supported catalysts methods has also been reviewed recently [81]. We observed the superior behavior of cobalt in catalysis, especially in electrocatalysis. Cobalt catalysts are very important due to their extensive applications in many industrial processes, such as Fisher–Tropsh synthesis [82,83] and CO_2_ conversion [84]. Electrocatalytic CO_2_ reduction reaction (CO_2_RR) is a promising strategy due to its easy operating system, simple constructions, operational at neutral pH, ambient temperature and atmospheric pressure, and low energy utilization to produce valuable chemicals and fuels such as formic acid, methane, ethanol, and carbon using renewable electricity. Therefore, CO_2_RR coupling with renewable energy sources can effectively achieve a carbon-neutral energy cycle and hydrocarbon products with high activity, stability, and selectivity [85,86].

Many expensive noble-metal electrocatalysts such as Au, Ag and Pd-based electrocatalysts have been employed for CO_2_RR, exhibiting high electrocatalytic performances, selectivity, and stability with low overpotential [87]. However, their wide use applications for CO_2_RR have been hampered by their high cost. Therefore, explorations to find the noble metal-free efficient catalysts for CO_2_RR have led to the discovery of various catalysts such as different types of metals, metal oxides, supported single atoms and single-site catalysts, transition metals supported/embedded on graphene, bimetal and metal/carbon hybrids, porphyrin like structures functionalized by metals, MOFs, colloidal nanocrystals and doped nanostructures [88,89,90,91,92,93,94,95,96,97,98,99,100,101]. Current challenges to the electrocatalysts are their stability, the reduction onset potential, current density, and Faradic efficiency. Another challenge is to find an electrocatalyst that is active for the CO_2_RR to produce useful products, while competing the hydrogen production reaction. The high reduction over potential leads to a waste of energy and a significant H_2_ evolution reaction (HER), a significant side reaction that prevails over the CO_2_RR.

Co-based materials have many advantages over others because as a popular metal, Co belongs to the group VIII-B of the periodic table having unique features like high electrical conductivity, thermal stability, unique electronic features, chemical stability, and high catalytic performances, which makes co-based catalysts as promising materials for CO_2_RR applications. Cobalt as an earth-abundant transition metal is a splendid alternative to noble metals such as Pt, Ir, Ru, etc. For CO_2_RR, Co has been used as a prominent source as noble metal-free electro/photocatalysts due to fascinating properties such as loosely bonded d-electrons and therefore readily available multiple oxidation states (Co(0), Co(I), Co(II), Co(III) and Co(IV). Moreover, it is found that a transition from Co(II) to Co(I) is involved at the intermediate state for CO_2_ reduction. Hence, high activity, outstanding stability and product selectivity are achieved through Co-based catalysts for CO_2_ reduction [84,102]. Cobalt is more reactive than other earth-abundant metals due to the possession of modest CO_2_ adsorption and d-band closeness to the Fermi level [103]. Co-based catalysts have been explored as effective cathode materials for electroreduction of CO_2_ to CO exhibiting high activities and selectivity [104]. For CO_2_ photoreduction, Co metal sites in Co-based MOFs offer the traps for electrons for facilitation in electrons-holes separation, thus providing a longer life for electrons for the reduction reaction. Co is found to be an important stabilizer for major intermediates in CO_2_ reduction [84,103]. Co-based materials have applications in various other fields such as energy storage, catalysis, and thermopower. Co-based materials (i.e., NaxCoO_2_) play a critical role in cathode and anode materials for Na-ion batteries. Likewise, LiCoO_2_ has been regarded as one of the most commercialized cathode materials for Li-ion batteries. Cobalt oxides and cobalt chalcogenides exhibit a high theoretical capacity for sodium storage [105]. Thus, cobalt has been reported as an important center for CO_2_ reduction [40,46,106,107,108,109]. 

The previous state-of-the-art review articles [84,110,111,112,113,114,115] have discussed ECO_2_RR more extensively. The ECO_2_RR using cobalt-based materials has gained considerable interest. Figure 1 shows the vast available literature on cobalt electrocatalysts and cobalt catalysts for CO_2_RR. Therefore, this review is designed to summarize cobalt-based electrocatalysts’ current progress in converting CO_2_ into valuable products. Emphasis is given to cobalt-based MOFs, COFs, cobalt complexes, cobalt oxides, cobalt nitrides, cobalt multi-metals and single cobalt atom catalysis.

## 2. Cobalt Catalysts for CO_2_ Reduction

### 2.1. Cobalt-Based Single-Atom Catalysts (SACs) for CO_2_RR

CO_2_ reduction reaction (CO_2_RR) catalysts have challenges of high overpotential, low Faradaic efficiency, low current density and lack of long-term stability. However, single-atom catalysts (SACs) are used for CO_2_RR with great importance. Studies show that SACs for CO_2_RR are of two categories based on their synthesis route: i) organometallic precursor pyrolysis such as MOFs; ii) loading of the metal precursor directly onto the support, which are followed by heat and acids treatment to get rid of the excess nanoparticles as shown in Figure 2. The Co precursor was dispersed with the polymer (Pluronic F127) and the colloidal silica, then the mixture was pyrolyzed and the template was etched by acid treatment to obtain the Co-SAS/HOPNC. Uniform hierarchical and atomic sites of cobalt dispersed in the carbon matrix were observed by the BET. SEM and TEM and the elemental analysis confirm the presence of the Co, N and C. 

The CO_2_RR was investigated by using a single-atom cobalt catalyst (Co-Typ-C) by Hou and coworkers [116] via the pyrolysis of Co-terpyridine (TPY) organo-metallic complex. The catalyst exhibited excellent acidity with CO Faradaic efficiency (FE) of >95% from −0.7–1.0 V (vs RHE). The catalysts without Co or Tpy ligand do not exhibit high CO FE. A virtual flue-gas with 15% CO_2_ was utilized as a source of CO_2_ to verify the catalyst’s efficacy, revealing that the CO FE was reserved at ~90% at a potential of −0.5 V (vs RHE). The CO partial current-density was promoted to 86.4 mAcm^−^^2^ and CO FE reached over 90% at a cell voltage of 3.4 V. The study indicates that a regular single atom Co-N_4_ site generally contributed to the good performance of the catalyst for CO_2_ reduction. A study by Shi et al. [117] shows that the atomic Co species (ACS) modified 2D terlliumnanosheet (Te Ns) act as an active single atom Co catalyst for CO_2_ reduction under visible light absorber. The ACS catalytic site serves as efficient electron transfer built via the coordinated Co center with five Co-O. Their finding reveals that the strong mutual interaction among Te Ns and ACS altered the electronic structure of Te Ns’, which induced the introduction of intermediate energy states acting as trap centers. The CO_2_ conversion reactions via the aid of [118] Cl_3_ complex as the light absorber gives the CO product amount of ~52.3 μmol/L, which has the best performance compared to the conventional catalysts under similar conditions. 

Huang and coworkers [120] fabricated partially-oxidized cobalt nanoparticles (5 nm) dispersed on a single-layered nitrogen-depend graphene catalyst (PO-5 nm Co/SL-NG) as shown in Figure 3. They revealed that the catalyst was efficient for selective CO_2_ electrocatalytic reduction to MeOH in a 0.1 M NaHCO_3_ electrolyte under mild conditions. A Faradaic efficiency of about 71% was reported for MeOH at −0.9 V (vs. SCE) and possessed a high electrocatalytic current density of 4 mAcm^−^^2^ with a yield of 1.0 mmoldm^−^^3^h^−^^1^. A low corresponding over the potential of ~280 mV was observed while Faradaiac efficiency for MeOH remain 23.3% at −1.0 V (vs SCE) at a current density of 10 mAcm^−^^2^. The catalyst was stable even after 10 h of the CO_2_ electro-reduction. They also propose a mechanism for CO_2_ reduction to methanol, as shown in Figure 3g. The support (SL-NG) promotes CO_2_ and multiple proton-transfer steps to produced methanol. Cobalt provides a synergistic effect between Co and supports to allow multiple proton- and electron-transfer steps to methanol in a high production rate. Gang et al. [121] used MOFs to obtain Co single-atom catalyst with four co-ordinates N (Co-N) and N/C on the N-incorporated porous carbon. XAFs results show that for binding N to single atom Co, the coordination number was dependent on pyrolysis temperature. Their findings show that the Co atom attached to four N in N-incorporated porous carbon (Co-N_4_) exhibit a Faradaic efficiency of about 82% with a current density of −15.8 mAcm^−^^2^ for electrochemical CO_2_ conversion to CO and was stable for 10 h. The mechanistic studies show that the Co-N_4_ active centers are promoted by the binding power of CO_2_, which facilitates the CO_2_ activation and is responsible for high-performance CO_2_ reduction. 

He et al. [122] investigated the use of a single metal atom supported on defective graphene as a catalyst for electrochemical CO_2_ conversion. The work studied the reaction pathway to produce C_1_, CO, HCHO, HCOOH, CH_3_OH and CH_4_ using the selected five transition metals (i.e., Ag, Co, Cu, Pt, and Pd). The work further reveals that the single-atom catalysts exhibit an entire enhancement in the rate-limiting potential to generate C_1_-hydrocarbons. However, a distinctive variation in terms of their efficiency and CO_2_ conversion selectivity was reported. These differences were correlated to their elemental behavior as a role of their group number in the periodic table for the CH_4_ generation. Thus single-atom catalysts are potential catalysts to overcome the overpotential problem associated with the CO_2_RR. 

### 2.2. Multi-Metals Cobalt Catalysts for CO_2_RR

Electrochemical CO_2_RR produces various forms of products ranging from CO, formic acid, alcohols, methane, olefin, and hydrocarbon [123]. However, the selectivity of these products is highly dependent on the adsorption characteristics of the reactants on the electrode surface. Due to the associated problems of metal electrocatalyst, transition metal catalysts could solve the problems to some extent because of the overpotential and poisoning [124], as metal and mixed metal electro catalysis still attracted scientific attention [125,126]. Recently, multimetallic compounds have been given more attention to reducing material cost and tuning the strength of intermediate electrochemical reaction of CO_2_ reduction to achieve high selectivity [127]. Multi-metals Co and Fe electrocatalyst were investigated for CO_2_-reduction by Abdinejad and coworkers [71,99]. Their research finding reveals that the introduction of amino substituent enhanced the electrolytic activity toward the CO_2_ conversion through dual active sites. The mono-amino FeP reduces the CO_2_ to CO at ambient temperature and pressure with significant turnover (TON). They further reported the reactivity and selectivity of amino compounds towards capture and electroreduction of CO_2_ in both homogeneous and heterogeneous [128]. 

Singh et al. [129] studied a direct electrochemical conversion of CO_2_ to valuable products, using non-noble bimetallic (Ag-Co) electrocatalysts. The role of the catalyst was investigated over the gas phase electrochemical CO_2_ conversion. It was found that, although the preferred metals are individually selective for CO generation, their combination results in the CH_4_ and C_2_H_4_ generation along with the CO. A maximum Faradaic efficiency of ~20% for CH_4_ production at 2 V using Ag-Co was reported. The effect of the binding power of the product formation during the CO_2_ reduction affects the applied potential, and chemisorption characteristics of the metal. Likewise, the product formation distribution which facilitates the C-C bond formation resulted in higher hydrocarbons.

The AFM images of multi-metals of Cu and Co nanoparticles (NPs) shown in Figure 4a–d have a homogeneous size ranging from 1.2–21 nm. The effect of these particles sizes was explored on CO_2_RR with different Co and Cu loading (i.e., x = 50, 70, 90 for Cu_x_Co_x-100_ system) by Bernal et al. [130]. The entire current density of the CuCo NPs as a function of nanoparticles size at −1.1 V vs. RHE presented in Figure 4e shows that the small-sized NPs are more active (i.e., activity increases with decrease NPs). The selectivity expressed in Figure 4f–h with a different nominal metal composition (Cu = 50%, 70% and 90%) shows an increase in H_2_ production with a decrease in NP size. The excellent activity was attributed to the presence of under coordinated site.

### 2.3. Cobalt Oxides Catalysts for CO_2_RR

Cobalt oxides have been used extensively in CO_2_RR as active catalysts as well as support materials. Aljabour et al. [131] used nanofibrous Co_3_O_4_ for the production of CO and formic acid. The nanofiber electrode exhibits stability for 8 h and overall Faradaic efficiency of about 90% for CO at a geometric current density of ~0.5 mAcm^−^^2^ on a flat surface shown in Figure 5.

Gao and coworkers [132] proposed an atomic layer for transition-metal oxide to improve the conventional electrocatalysts that frequently suffer from low efficiency and weak durability. The catalyst offers an ultra-large fraction of the active sites, elevated electronic conductivity and fine structural stability. Atomic thickness (1.72 and 3.52 nm) endowed the Co_3_O_4_ with abundant active sites that ensured a large CO_2_ adsorption. The Co_3_O_4_ layer with a thickness of 1.72 nm displayed over 1.5 and 20-times superior electrocatalytic performance than the 3.51 nm-thick layer electrode. The 1.72 nm Co_3_O_4_ layer displayed formate Faradaic efficiency of >60% in 2 h. The Zn and Co_3_O_4_ on graphite-plate works as the cathode and anode, respectively in the presence of Na and K carbonate and bicarbonate electrolytes. Their finding shows that HCOOH was the selective resultant product under all the applied conditions. The highest efficiency was achieved with the bicarbonates, using the KHCO_3_ electrolyte at 1.5 V, maximum Faradaic efficiency for HCOOH (78.5% and 78.46%) were achieved after 5 and 10 min reaction time. Similarly, 50% and 64.7% efficiency for CO was observed with NaHCO_3_ electrolyte at 2.5 V after 5 and 10 min, respectively. Similarly, another study by the same authors [133] for electrochemical reduction CO_2_. The study was conducted using CO_2_ dissolved in 0.5 M KHCO_3_ electrolyte at diverse applied voltages (1.5–3.5 V) and at a time interval of 5–25 min. The CO_2_ was converted to various products (such as HCOOH, CH_2_O, CH_3_H_8_O, CH_3_COOH, CH_3_OH) and ethanol as the main product. The ethanol as the main product of CO_2_ reduction with maximum Faradaic efficiency of 76.31% and 96.15% at 1.5 and 2 V, respectively, was obtained in a 5 min reaction. Cheng et al. [134] incorporated metal-oxide nanoclusters such as FeOx, NiOx and CoOx into the iron phthalocyanine (MOx/FePc) supported on graphene as electrocatalysts. The catalysts displayed high activity and selectivity along with stability for electrochemical CO_2_ reduction. Their findings show that the excellent activity is due to the fact that the MOx/FePc catalysts undergo replacement of the metal-ion with the iron center of Pc, thereby producing an electrochemical substituted metal-Pc, (i.e., e-MPc, M = Co and Ni) co-existing with the substituted FeOx NPs in the neighborhood of e-MPc. Guo et al. [135] introduced the concept of combining an H_2_ evolution reaction (HER) with the CDots/C_3_N_4_ heterojunction, a CO_2_ reduction electrocatalyst to have a cheap, stable, efficient and selective route for excellent production of syngas. They utilized MoS_2_, Co_3_O_4_, Au and Pt as the HER components. The Co_3_O_4_-CDots-C_3_N_4_ electrocatalyst was highly efficient and the produced syngas H_2_/CO ratio was tuned from 0.07:1 to 4:1. The catalyst was found to be excellent for syngas production as the only product for >100 h. Gao et al. investigated the roles of oxygen vacancies in CO_2_ electro reduction with an apparent atomic-level association between the oxygen and the CO_2_ electro-reduction. The studies constructed model oxygen vacancies confined in an atomic layer using oxygen-deficient cobalt oxide single-unit layers. It was shown that the existence of oxygen (II) vacancies lowered the rate-limiting activation barrier from 0.51–0.40 eV via the stabilized formate-anion (HCOO-) radical intermediates. This was confirmed via the reduced onset potential from 0.81–0.78 V and the declined Tafel slope from 48–37 mVdec^−^^1^. The vacancies riched cobalt oxide single unit cell layer displayed a current density of 2.7 mAcm^−^^2^ with formate selectivity of 85%, during the 40 h test. Aljabour et al. [131] studied the catalytic behavior of semi-conducting Co_3_O_4_ nanofibers catalyst for CO_2_ conversion to CO using 65% Faradaic efficiency. The studies demonstrated the use of Co electrocatalysts without using any additional metal by expanding the electrode network with Nano-fibrous interconnection. A polyacrylnitrile polymer template was used to fabricate highly ordered Co_3_O_4_ fibers to expand the catalyst’s surface-to-volume ratio. The nano-fibrous electrode was stable for 8 h at a current density of ~0.5 mAcm^−^^2^ on a flat surface. A 65% Faradaic efficiency for CO_2_ conversion to CO was reported with the Co_3_O_4_ nanofibers electrode. The amount of the products formed were increased by increasing electrolysis time. It was observed that CO formation dominates the format production which is attributed to the low proton number in the electrolyte. It was found that the optimum limit for operating voltage, the utmost Faradaic efficiency for CO generation was achieved at −1560 mV vs. NHE. The nanofibrous Co_3_O_4_ electrodes performed for 8 h at a current density of 0.5 mA/cm^2^. Significant degradation of the nanofiber electrode was observed but still stable and operational. 

The bimetallic cobalt oxide with other metals has also been explored for ECO_2_RR. The tin (Sn) and cobalt oxide (Co_3_O_4_) electrocatalytic effect for reduction of CO_2_ was investigated by Yadav and Purkait [136]. The process was performed by using Sn and Co_3_O_4_ as cathode and anode, respectively, in 0.5 M KHCO_3_ and NaHCO_3_ electrolyte solutions at various applied voltages ranges and times. Their findings show formic acid formation with high Faradaic efficiencies of ~74% and 93% for the CHOOH product obtained from 20 and 5 min reactions at 1.5 and 2 V applied potential. A CO_2_RR to formate with low overpotential utilizing the Co_3_O_4_-CeO_2_/low graphitic carbon (LGC) catalyst with oxygen-vacancies was studied by Zhang et al. [137]. The studies reveal that the overpotential for CO_2_RR to produce formate on the Co_3_O_4_- CeO_2_/LGC electrocatalyst was ~0.31 V (vs. RHE) and the highest Faradaic efficiency of the format was 76.4% at −0.75 V vs. RHE (−61 mAcm^−^^2^) with high stability (i.e., 45 h). The catalyst displayed the formate production rate of 1.6 mmolm^−^^2^s^−^^1^C^−^^1^mg^−^^1^. It was shown that the coupling of Co_3_O_4_ to CeO_2_/LGC led to a high concentration of oxygen vacancies and is regarded to be a key in the enhanced performance and stable selectivity with inhibited H_2_-evolution reaction. A beached Au@Co_3_O_4_ nanowire array (Au@Co_3_O_4_ NAs) obtained by immersion in HAuCl_4_ solution with Co_3_O_4_ nanowires was studied by Zhang et al. [138]. The catalyst exhibited excellent electrochemical activity by generating syngas (H_2_ and CO) via turning from 1:1–4:1 with controlling voltage. It also showed a better catalytic activity for the oxygen evolution reaction with an overpotential of 320 mV at 50 mAcm^−^^2^ current density and a small Tafel slope of 75 mVdec^−^^1^ in 1 M KOH electrolyte. The electrochemical activity of Au@Co_3_O_4_ NAs was better than the Co_3_O_4_ NAs, which opens up a feasible approach for the design and construction of a bifunctional electro-catalyst for an overall reaction. A proto-type bifunctional Co_3_S_4_@Co_3_O_4_ core–shell octahedron heterojunction catalyst synthesized by Yan and coworkers [139] was used for electro-reduction of oxygen and CO_2_ with high activity. The bifunctional electrocatalyst was shown to consist of a p-type core Co_3_O_4_ and n-type shell Co_3_S_4_ with high electron density. The synergistic interaction between the Co_3_S_4_ and Co_3_O_4_ bi-catalyst reduced the activation energy for conversion of the intermediates and enabled the CO_2_-reduction reaction at low potentials. Formate was the highly selective main product at high Faradaic efficiency of ~85%, accredited to the synergistic coupling effect of co-catalyst structure. The exceptional synergistic effect of the metallic Co and the encapsulated coordinately unsaturated CoO display high activity for clean production of CO under moderate conditions. A CO_2_ conversion of ~19% was attained with N-incorporated defective CoO shell (Co@CoO-N) with an exceptional CO production rate of 96 mLmin^−1^g_cat_^−1^ at 523 K with 42,000 mLg_cat_^−1^h^−1^ GHSV. The reaction mechanism is also described in detail as shown in Figure 6. Briefly, the surface of the Co-S bond adsorbs CO_2_ to generate CO_2_•− intermediate. The Co_3_O_4_ favors water splitting to form H^+^ and OH^−^. Subsequently, the CO_2_•− reacts with H+ and e− to create the product.

### 2.4. Cobalt-Based Nitride Catalysts

Cobalt nitrides have proved beneficial for CO_2_RR because they offer more surface base sites of adsorption of CO_2_ and the generation of more active sites [140,141,142,143,144,145,146]. Peng et al. reported the highly efficient CO_2_RR cobalt nitride catalysts (700-Co_5.47_N/C) in an aqueous electrolyte. For the synthesis of these cobalt nitride catalysts, impregnation and nitridation involving temperature-programmed reaction (TRP) were used. For the optimized electrocatalyst 700-Co_5.47_N/C, the observed CO current density was 9.78 mA. cm^−^^2^ at −0.7 V vs. RHE. The as-prepared electrocatalyst showed high Faradaic efficiency and good stability. Furthermore, tuning of the electrolysis potentials led to the CO/H_2_ ratio adjustment from 3:1 to 3:2 [44]. In 2019, a robust synthesis strategy was adopted to synthesize the metal/nitrogen/carbon (MNC) catalysts with the presence of metal atoms as the atomically dispersed metal-N_x_ moieties (wherein MN_x_, M represents Mn, Co, Fe, Ni, and Cu metals) in N doped carbon using Zn MOF and metal salts. Jaouen et al. identified a volcano trend in MNC catalysts with MN_x_ (M = Mn, Co, Fe, Ni, and Cu), providing an in-depth understanding of the activity and selectivity of atomically MN_x_ with different metals for CO_2_RR. MNC catalysts as promising candidates for CO_2_RR were studied as model catalysts. Co was observed at the top of the volcano based on electrochemical potential. The experimental-operando X-ray absorption near edge structural spectroscopy was used for accurate modelling of active sites keeping the changes in the oxidation states of metals with change in potential. CoNC had no change in oxidation state with changing potential, and M2 + N4-H_2_O was identified as the most active center by computational studies. This work provided a base for the design and fabrication of cobalt-based MNC catalysts for CO_2_RR to be used in the future [45]. The cobalt nitrogen functionalized materials are noteworthy catalysts for CO_2_RR due to their high-performance activities. Metalloporphyrin and its derivatives have also been promising materials for catalytic activities [14]. Therefore, in 2019, Zhou et al. employed DFT calculations to study CO_2_RR on Co-centered porphyrin and graphene with C, N and O as different coordinating atoms to further improve the activities. Through coordination engineering, the catalytic activities can be enhanced by the cobalt atom’s vacancy formation energy arising by substituting different coordinating atoms. Detailed electronic studies results showed that Co-O bonds lack π -bonding compared to the Co-N and Co-C bonds in the Co-centered structure, therefore, had potential for high catalytic activities. Hence, coordination engineering can be employed as an effective strategy for the enhancement of CO_2_RR catalytic activities in cobalt nitrogen functionalized materials [147].

The CO_2_RR performance of Co can be tuned via altering its coordination environment [121]. In 2020, Wang et al. utilized metal–organic layers (MOLs) as 2D analogous materials to MOFs and tuned the N-atom coordination number from 2 to 5 in the first sphere around the Co centers present on MOLs for optimization of electrocatalytic CO_2_RR as shown in Figure 7. In these catalysts, Co centers supported on MOLs were chelated by the Bipyridine and terpyridine sites present on MOLs which prevented the equal distribution of ligands or other dimerization processes. Following phenylpyridine or bipyridine introduction to the chelated Co centers supported on MOLs led to the formation of CoN_4_ and Co- N_5_ centers. Therefore, in the as-prepared catalysts MOL-Co-N_x_ (x = 2 to 5), bipyridine, phenylpyridine, and terpyridine ligands are the sources of different N atoms. In the case of MOL-Co-N_3_, wrinkled ultrathin film-like structures are observed through TEM. Meanwhile HAADF and HRTEM depicted hexagonal structure patterns, which confirmed the Hf_6_ SBUs present in MOL-Co-N_3_. Among all the as-synthesized electrocatalysts, MOL-Co-N_3_ displayed the highest CO_2_RR performance with a significantly high CO Faradaic efficiency (FE-CO) of 99% at −0.5 V_RHE_ [43]. The development of appropriate co-catalysts along with photocatalysts is a promising strategy for boosting CO_2_ reduction. The use of co-catalyst has various advantages including lowered activation energy for CO_2_, increased selectivity, promoted surface charge separation and enhanced stability. Therefore, keeping in view these advantages, metal nitrides can prove to be beneficial candidates for co-catalysts. Co-based nitrides can serve as an effective material to act as co-catalyst. Therefore, Liu et al., keeping in view the properties of metal nitrides, specifically cobalt nitrides, produced a novel cobalt nitride as a noble metal-free co-catalyst for enhancing the performance of BiOBr ultrathin nanosheets for effective CO_2_ reduction. For the synthesis of Co_2_N, a Co-based precursor was subjected to nitridation treatment and then Co_2_N/BiOBr hybrid was formed. Exploration through TEM, aberration corrected-scanning TEM with high angle annular-dark-field imaging (HAADF-STEM) and atomic-resolved STEM image revealed the transparent morphology of BiOBr with an ultrathin thickness of 1.6 or 2.5 nm for 2 to 3 layers and lattice fringes of 0.277 nm corresponding to (110) plane. The morphology of Co_2_N exhibits irregular structure with STEM images demonstrating the orthorhombic Co_2_N. Meanwhile, coupling of BiOBr and Co_2_N showed a well-defined nanosheet structure with uniform particle distribution. High selectivity CO formation rate was observed for Co_2_N/BiOBr hybrid approximately 6 times higher than BiOBr with lowered activation energy and high stability leading to optimized CO_2_ reduction activity [148]. 

For further insight into the enhancement of catalytic performances of TMNs for CO_2_ RR, theoretical studies were presented for a clear understanding. Recently, Karamad et al. evaluated the catalytic activity and stability of various TMNs and their hetero-doped structures for CORR through high-throughput density functional theory (DFT). The results presented that various TMNs can activate the CORR but have electrochemical stability problems, including the competitive H_2_ reaction. Therefore, only a few TMNs were found to be stable and active electrocatalysts [149]. The above discussion highlights the importance of cobalt nitrides for enhanced CO_2_RR. These research works presented efforts to optimize the activity and selectivity of CO_2_RR by tuning the electronic structures, compositions, morphology, and coordination chemistry of cobalt nitrides using different precursors and synthesis strategies to eliminate the drawbacks to achieve excellent activity and stability for CO_2_RR.

### 2.5. Cobalt-Based Complexes for CO_2_RR

Transition metal complexes offer an advantage for CO_2_RR due to fine-tuning the coordination sphere via altering the chelating surroundings vis-à-vis electronic and steric effects of the chelating agents. Such fine-tuning is not possible in solid-state transition metal catalysts. Metal complex-type catalysts are available in the literature, ranging from noble metals (Ir, Ru, Re, etc.) to none-noble coinage metals (Co, Ni, Fe Cu, etc.) [150,151]. These metals provide a two-electron reduction pathway to form -COOH, using organic reaction media. The porphyrin ring is an efficient ligand among other ligands because of its peculiar stability and high photo-electrochemical traits. A variety of cobalt complexes have been investigated for CO_2_RR with promising results, there is still a need to find and generate a low-valent intermediate with significantly lower potential.

Qiu Tian reported CO_2_ reduction over Ni and Co macrocyclic metallic complexes. In their findings, the tetramethylated Ni(TMcyclen) complex exhibited the best Faraday efficiency for CO production, which was followed by dimethylated Ni(DMcyclen) complex, and the Ni(cyclen) had no catalytic effect on the reaction [152]. Wang et al. reported CO_2_ electrochemical reduction for cobalt phthalocyanine complexes supported over carbon (CoPc@carbon). The phthalocyanine used one trimethyl ammonium side chain and three tert-butyl moieties on the phthalocyanine macrocycle. They obtained a high current density of 165 mA.cm^−^^2^ for CO against a potential of 0.92 V vs. RHE as shown in Figure 8. There highest CO selectivity was 99%, and stability for around 3 hr. the CoPc@carbon complexes were investigated using XANES spectra, which revealed the typical features of Co(II) phthalocyanine complexes [65]. Ogawa et al. reported CO_2_ reduction over cobalt monoanionicbipyricorrole (Co(II)BIPC) complex and reported less overpotential. The crystal structure of Co(II)BIPC was determined by crystallography. They did cyclic voltammetry in DMF, which revealed two reversible peaks, i.e., E1/2 at −0.87 V and −1.75 V. they further corroborated that the first reduction even at −0.87 V is due to the Co(II)BIPC conversion to Co(I)BIPC. They conducted CO_2_-to-CO electrocatalytic reactions under the CO_2_ atmosphere, with emphasis on reducing the overpotential needed for the reaction. There Faraday efficiency was 75% at the overpotential of 0.35 V [69]. Roy et al. reported CO_2_ reduction over cobalt phospho complexes [CpCo(PR2NR′2)I] having diphospine ligands and pendant amine moeities. The diphospine and pendent amine residues were claimed to provide the necessary proton relay redox system for CRRR. They claimed over 90% selectivity for formic acid at 0.5–0.7 V overpotential in DMF. They found that the Co complexes having the highest basic amine and the most electron-donating diphosphine moieties showed the best catalytic properties, i.e., TOF > 1000 s^−1^. They corroborated their catalyst mechanism with the first-principle calculations employing TURBOMOLE, after geometry optimization with B3LYP functional. Their finding revealed the simplest possible way to convert CO_2_ to formic acid, provided that the catalyst stability was properly addressed [64]. Nganga et al. [66] prepared a series of mono-, di-, and tri-β-oxo-substituted porphyrinoid-cobalt(II) complexes from porphyrin, for electrocatalytic conversion of CO_2_. They reported that the monooxochlorin complex 2Co exhibit highest current than the dimonooxochlorin 6Co, 7Co and dioxobacteriochlorin 4Co. in there report thy further reported the production of Co(ii) complex–CH_4_ via β-oxoporphyrinoid complexes, whereas CH_4_ production was enhanced by β-oxoporphyrinoid complexes by 8 electron redox cycle. Xia et al. [67] reported on the dual active sites in cobalt phthalocyanine complexes for CO_2_R. They studied the reaction mechanism using synchrotron-based XAS and XPS analysis. They claimed that the catalyst showed dual active sites for the CO_2_RR. in the first step, CO_2_ gets attached to the N site on the cobalt phthalocyanine complex via protonating CO_2_, to form a -COOH intermediate. In the second step, -COOH gets transferred to the Co central metal of the complex and is reduced to CO at the workable potential. By applying an optimum potential of 0.8 V, they reported over 95% CO production for a stable period of 5000 s. Liu et al. [153] used polymer coordination and encapsulation strategy, by using poly-4-vinylpyridine. Their catalytic system found >92% selectivity for CO over H_2_ for 2 h, at −1.25 V vs. SCE and −2.90 mA.cm^−^^2^ current density. The authors used kinetic isotope and proton inventory effect to explain the evident boast in the activity and selectivity of their polymer encapsulated cobalt phthalocyanine complexes. They reported that the proton relay mechanism is responsible for the higher selectivity towards CO, which suppressed the hydrogen evolution reactions. To boost the electrochemical properties of cobalt phthalocyanine complexes, Ma et al. [68] used polymer coordination and encapsulation strategy by using poly-4-vinylpyridine. Their catalytic system found >92% selectivity for CO over H_2_ for 2 h, at −1.25 V vs. SCE and −2.90 mA.cm^−^^2^ current density. The authors used kinetic isotope and proton inventory effect to explain the evident boast in the activity and selectivity of their polymer encapsulated cobalt phthalocyanine complexes. They reported that the proton relay mechanism is responsible for the higher selectivity towards CO, which suppressed the hydrogen evolution reactions. Jin et al. [154] studied the Mechanism of the CO_2_RR by cobalt Porphyrins towards CO, Formic acid, and methane. They report a density functional theory (DFT) study as shown in Figure 9. According to this finding that CO_2_^−^ anion the key intermediate, was formed when cobalt exists in the Co^I^ oxidation state in cobalt porphyrins complexes. The CO generator is a major product during a decoupled proton-electron transfer. Methane has the lowest Faradic efficiency via subsequent CO conversion by the concerted proton-coupled electron-transfer reactions. Similarly, the formic acid product was minor through a [Co(P)–(OCHO)] intermediate. 

### 2.6. Cobalt Porphyrin for CO_2_RR

In recent years, porphyrin-based metal complexes have been significantly explored for CO_2_ reduction [155]. This section keeps separately due to its broad research because porphyrin-ring offers high electron transfer activity, thus requiring relatively lower overpotential with high CO selectivity in the CO_2_ reduction reactions. Additionally, researchers have made significant progress in porphyrin-based electrocatalysts anchored on potential host materials in terms of catalyst stability. Immobilized cobalt porphyrins have been reported for high faraday efficiency towards CO with lower overpotential. First principle calculations have also played a pivotal part in explaining the electronic structures of the metal-porphyrin catalytic systems for CO_2_ reduction [156]. Shen et al. [70] immobilized cobalt protoporphyrins on pyrolytic graphite in aqueous media, and tested them for CO_2_ reduction. The authors underpinned the pH stability of cobalt porphyrin electrocatalyst, saying that optimization of pH plays a crucial role in triggering the CO_2_ reduction reaction. Their study claimed a >60% Faraday efficiency towards CO and >2.5% towards CH_4_ was reported at a potential of −0.6 V vs. RHE, under acidic conditions. First principle calculations are important to understand the reaction pathways in the cobalt porphyrin electrocatalytic systems. In this regard, Leung et al. [157] performed the pioneering work by applying quantum chemistry and ab initio calculations in investigating the electrocatalytic effect of cobalt porphyrin complexes in the reductive decomposition of CO_2_ in aqueous media. They used hybrid functionals alongside the dielectric continuum solcation methods, for determining the electron transfer mechanisms. They proposed a reductive mechanism that explained the CO_2_ reduction at ambient to higher pH levels, which starts with one-electron transfer to the cobalt porphyrin complex [Co-porphyrin]^−^, which then adds CO_2_ to itself forming adjunct an [Co-porphyrin.CO_2_]_2_^−^. Upon protonation an intermediate [Co-porphyrin.COOH]^−^ is produced, which upon relieving the OH^−^ moiety, produces the product CO. The two key intermediates in this whole reaction cycle are [Co-porphyrin.CO_2_]_2_ and [Co-porphyrin.COOH]^−^, due to the stronger interactions of CO_2_ and water. Miyamoto and Asahi [158] reported on the enhanced CO_2_ reduction to CO over cobalt porphyrin complexes in water based upon Koper’s mechanism. They combined their DFT calculations with the experimental results. Their work targeted the pH stability of their catalytic system, and reported that at pH = 3, a single proton shift from water occurs at −0.8 V vs. RHE. They said that water plays a vital role in the CO_2_ reduction to CO, as their calculated redox potential of proton transfer matched that of the experimental results. 

As a rule of thumb, the electron-withdrawing moieties on the electrocatalyst will facilitate the shifting of electrons to the catalyst core, but at the expense of the proton transfer step. Similarly, the electron-donating moieties facilitate the shifting of protons at the expense of the electron transfer step. Therefore, it is needed to find a middle ground between the electron-donating and electron-withdrawing moieties on the catalyst, to harness the CO_2_ reduction reaction as per our needs. In this regard, amino groups attached to the porphyrin ring may come useful because the -NH_2_ species acts as an electron-donating moiety that reverses its role to the electron-withdrawing group upon protonation. Abdinejad et al. [71] prepared cobalt and iron porphyrin complexes, where the porphyrin ring had amino moieties as electron transfer boasters. The authors attached different amino moieties to the porphyrin ring and reported higher catalytic activity of their catalyst. In their findings, cobalt porphyrin complexes were suitable for hydrogen production, while iron porphyrins were the most suitable catalysts for CO production. The FE selectivities towards CO and H_2_ for the cobalt porphyrin were 8% and 99%, respectively. Meanwhile, the iron porphyrin produced 49% CO and 50% H_2_ in their aqueous electrochemical catalytic system (Table 1).

### 2.7. Cobalt-Based MOFs and COFs for CO_2_RR

The reticular chemistry of the MOFs and COFs enable the use of tunable control of the catalytic system to convert the carbon dioxide to value-added products [159]. However, this tunable property is prohibited by poor electrical conductivities. This can only be overcome by the suitable use of the metals such as cobalt, copper in the inorganic SBUs.Katherine A. Mirica reported the synthesis of conductive two-dimensional MOF made of metallophalocyanine (cobalt/nickel) ligands linked by copper nodes with high electrical conductivities for the conversion of CO_2_ to CO. It was observed that TOF values range from 1.15 and 0.63 s^−^^1^ for CoPc-Cu-NH and CoPc-Cu-O [160]. Christopher J. Chang reported COF synthesis containing cobalt porphyrin catalysts as building units and organic linkers bonded through imine linkage to make the COF for the aqueous electrochemical reduction of CO_2_ to CO. The COF materials were deposited on porous, conductive carbon fabric. Incorporating tubular molecular units of the porphyrins within the extended COF structure gave an advantage in electrocatalytic reduction with exceptionally high activity and selectivity. Thus we find that with the increasing length of the linker from COF-366 to COF-367, the activity is increased with the TON up to 290,000 and TOF of 9400 h^−^^1^ about 26-fold more than the normal cobalt complex with no degradation over 24 h [161]. Rong Xu immobilized cobalt oxide nanoparticles on MIL-101 for water oxidation with a TOF of 0.012 s^−^^1^ [162]. Jinhong et al. prepared cobalt immobilized on a covalent triazine-based framework (CTF) as an efficient cocatalyst to reduce CO_2_ under visible-light irradiation. The CTF helps in the CO_2_ adsorption while the pore structure helps in the accommodation of CO_2_ and electron mediator. It was observed that the production of CO increases 44-fold than the pristine CTF on the introduction of cobalt. The obtained CO from this catalyst was about 50 mol g^−^^1^h^−^^1^ [163]. Peidong Yang et al. prepared a cobalt porphyrin-based Aluminum MOF for the conversion of CO_2_ to CO with the TON up to 1400. In situ analysis showed that the majority of the redox-accessible Co(II) is reduced to Co(I) during catalysis [164]. Xinyong Li prepared a novel Z-scheme heterojunction of Co_3_O_4_@CoFe_2_O_4_ hierarchical hollow double-shelled nanoboxes derived from ZIF-67 to give CH_4_ and CO at a rate of 2.06 mol h^−^^1^ and 72.2 mol h^−^^1,^ respectively [164]. Yaghi et al. [165] prepared a new anionic 3D metal–organic framework MOF-1992 containing Co phthalocyaninoctaol. It converts CO_2_ to CO with the TON up to 5800 and TOF 0.20 s^−^^1^ with a current density of −16.2 mA cm^−^^2^ at −0.52. The electroactive coverage of the catalyst was estimated with the aid of the CV measurement (Figure 10), and it was found to be ~25% of the total catalyst loading. Zhang et al. [166] studied a novel mixed-metallic MOF [Ag_4_Co_2_(PYZ)PDC_4_] which transformed into an Ag-doped CoO_4_ catalyst. The Ag/Co_3_O_4_ catalyst gives the highest selectivity for CO in 0.1 M KHCO_3_ electrolyte (CO_2_ saturation), up to ~45% Faradaic efficiency was reported. Compared to the Ag/Co_3_O_4_ electrode, the highest Faradaic efficiency for CO over the pure Co_3_O_4_ is 21.3% at 1.8 V (vs. SCE). Their findings show that the presence of Ag improves the efficiency of CO significantly and inhibits H2 production for 10 h at −1.8 V (vs. SCE). Pan et al. [40] used cobalt MOF as a precursor to synthesized electrocatalysts on carbon as model catalysts for CO_2_RR. The prepared Co electrocatalyst was compared with Fe MOF-based electrocatalysts. The MOFs-derived catalysts were more active than the bulk electrocatalysts for CO_2_ reduction over hydrogen production reaction.

## 3. Summary and Outlook

Carbon dioxide mitigation by non-noble, earth-abundant and cost-efficient catalysts like cobalt is favored for the long run. Due to several structural characteristics like unsaturated d-orbitals, multivalency, high coordinating ability, it can form a variety of catalysts. In addition, these cobalt catalysts have moderate CO_2_ adsorption, high surface area and availability of more active sites for CO_2_RR. Similarly, the latest research on cobalt catalysts, focusing on exploration through experimental studies and theoretical studies and emphasizing design and fabrication of cobalt-based electrocatalysts for CO_2_RR, were summarized. Thus, this review’s critical analysis and guidance will provide new insights to future research and best practices for CO_2_RR.

The electrochemical reduction of CO_2_ using single-atom catalysts to generate high-value products is an attractive means of addressing the effects of alarming CO_2_ emissions and the associated problems. Appropriate and increasing successes recorded in this field will address the ever-presented societal demand for chemicals and fuels. There is a need to develop excellent catalyst chemistries for this reaction, and it is always required to tune the activity and selectivity of the process, especially for the increased complex chemistries, such as in the oxides of cobalt. Determining their surface binding energies towards the CO and H_2_ adsorption using density functional theory will be crucial for evaluating their activity and selectivity. The use of multi-metallic catalysts is very active for the direct electrochemical reduction of CO_2_ into valuable hydrocarbons. This is very important from the point of view to generate fuels from the CO_2_. The formation of the hydrocarbons resulted from the alteration of binding energies and modified chemisorption properties of the involved catalyst metal. This is usually helpful in the development of a more effective and selective electro-catalyst. Cobalt complexes have great electronic conductivity and high Faradic efficiency. The porphyrin ring is an efficient complex ligand among several ligands because of its peculiar stability and high photo-electrochemical traits. A variety of cobalt complexes have been investigated for CO_2_RR with good current density and Faradic efficiency but there is still a need to generate low-valent intermediate with significantly low overpotential. MOFs and COFs showed promising results for CO_2_RR. However, their product Faradic efficiency is limited to CO and COOH only. This further enhancement in reaction product to higher alkanes is needed. Co single-atom catalysts which are obtained from the pyrolysis of cobalt-containing organometallic complexes, polymer, MOF or by the loading of cobalt precursor onto the support, were revealed as promising catalysts for the CO_2_RR. The single uniform atom leads to strong binding with the CO_2_ resulting in an efficient reduction reaction with high Faradic efficiency toward CO formation. One of the few studies that showed the conversion of CO_2_ to MeOH uses partially oxidized 5 nm cobalt nanoparticles dispersed on single layer nitrogen-depend graphene catalyst (PO-5 nm Co/SL-NG) with Faradaic efficiency of ~71% and current density of 4mAcm^−2^ at 0.9 V. 

A number of cobalt oxides were tested for the CO_2_RR, nanofibers, atomic layer oxide, Zn-Co_2_O_3_ and Co_2_O_3_-FePc revealed decent Faradic efficiency for CO and HCOO-. Several factors affect the CO_2_RR performance, selectivity and stability, the atomic thickness, which provides large actives sites for the CO_2_ adsorption. The presence of oxygen vacancies in Co increases the performance and selectivity toward HCOO- formation. 

Different C1 and C2 products (with ethanol as the main product) were also investigated upon the use of Co_3_O_4_ as anode electrode with Faradic efficiency of 76.31% and 96.15% at 1.5 and 2 V respectively were obtained in 5 min reaction. Meanwhile, H_2_/CO was formed in a ratio of 0.07:1 to 4:1 by using The Co_3_O_4_-CDots-C_3_N_4_ electrocatalyst. Several studies investigate the utilization of cobalt nitride-based electrocatalysts for CO_2_RR. With the use of various synthesis strategies, the coordination environment and state of Co, N and O can be tuned, which significantly influences the reduction performance. Experimental and computational data showed M2 + N4-H_2_O and Co-Nx (x = 2–5) are the most active center sites for CO_2_ reduction reaction and exhibit excellent Faradic efficiency for CO formation. Moreover, Co-based nitrides can be considered as an efficient co-catalyst for the photoelectrochemical reduction of CO_2_. Co_2_N served as a co-catalyst for the photocatalyst BiOBr and a remarkable enhancement in the selectivity, lower activation energy and stability toward the CO formation.

Among different cobalt complexes, cobalt-based porphyrins have been wildly explored for CO_2_RR. The mechanistic studies explain that upon reduction reaction, the electron is attracted to the metal center, followed by the attachment of CO_2_ to the complex, to be protonated to form the intermediate (Co-porphyrin.COOH), to produce CO in the presence of the electrolyte (OH-). Additionally, the amine moiety in the porphyrins serves as an electron-donating group, which facilitates the electron transfer to the complex center. However, some cobalt porphyrins with amines groups showed selectivity toward H_2_ production compared to iron complexes.

Despite several advancements, further improvements for recommendation are given below:Mixed metal catalysts: The numbers of electrons involved in the ECO_2_RR are 2,8,12,14 for the formation of CO, methane, ethylene, and ethane, respectively. Bi and trimetallic catalysts are required to allow the charge transfer for the desire products especially CO_2,_ towards higher hydrocarbons.Complexes: The use of organometallic complexes is still needed to increase the conjugation of the materials to overcome the conductivity problem for CO_2_RR for products higher than the eight-electron system.Nanostructured electrocatalysts: The use of nanostructured cobalt catalysts can be further improved to gain a high current and to provide a high density of active sites for CO_2_RR.In situ/operando measurements: It is crucial to understand the catalyst’s active sites, reduction pathway, and the type of intermediates by operando techniques.Simulation data: The CO_2_ electro reduction can be combined with simulation electrocatalysts to predict the product efficiency. A better reaction is needed to predict the catalyst development for efficient CO_2_RR.Cobalt sulfide and selenide: The composites of cobalt sulfide and selenide can be explored for electrochemical CO_2_RR.CO_2_ reduction under impurities: The post-combustion plant contains several impurities along with CO_2_. Thus CO_2_RR in the presence of impurities must be tested to revealed electrode activity and stability.Mixed gases data: CO_2_ in the air is present as a mixture with other gases. The practical application of this technology is required to reduce the CO_2_ in mixed gases electrochemically.Current density: Currently, cobalt-based materials used liquid electrolytes with bubbled CO_2_ for reduction. However, this gives low current density. The gas diffusion cells can be introduced to overcome this solubility issue by the liquid electrolytes and to achieve high current densities (>100 mA).Use of solid electrolytes: A solid-state electrolyte configuration might overcome the challenge of CO_2_ solubility and low current density.

## Figures and Tables

**Figure 1 nanomaterials-11-02029-f001:**
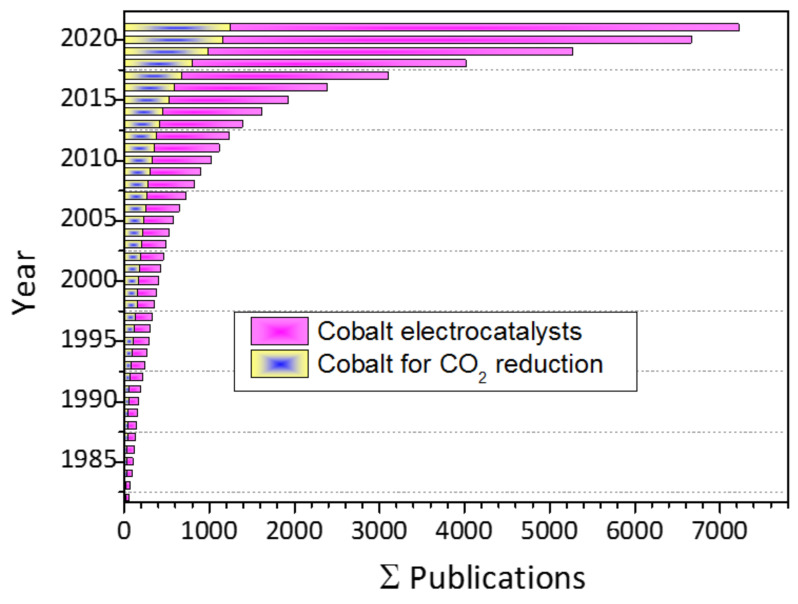
Published literature on cobalt electrocatalyst and cobalt catalysts for CO_2_ conversion. Source SciFinder, keyword: cobalt electrocatalyst, cobalt catalyst for CO_2_ reduction.

**Figure 2 nanomaterials-11-02029-f002:**
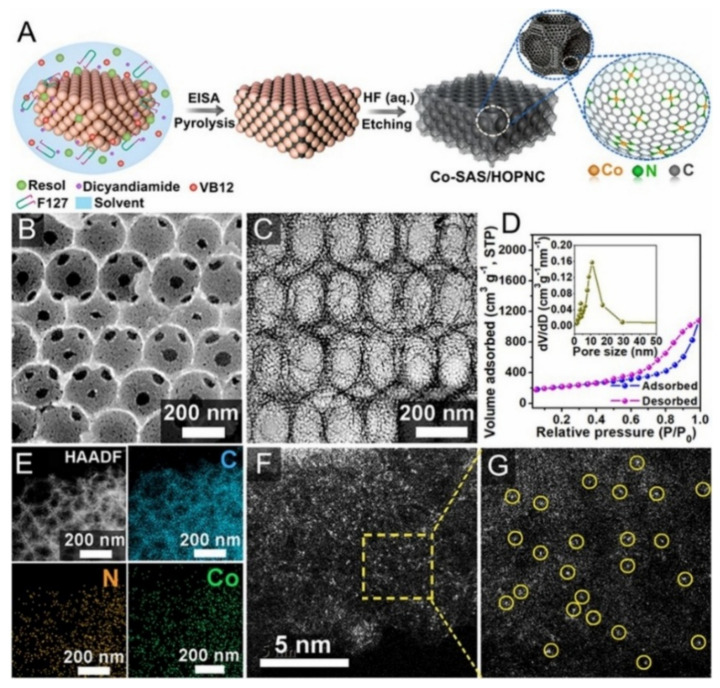
Fabrication of Co-SAS/HOPNC catalyst. (**A**) Step-wise synthesis procedure of Co-SAS/HOPNC. (**B**) SEM image, and (**C**) TEM micrograph of Co-SAS/HOPNC catalyst. (**D**) N_2_ adsorption–desorption isotherm curves of Co-SAS/HOPNC catalyst. (*Inset*) The pore size distribution curve. (**E**) HAADF-STEM micrograph and the EDS maps of Co-SAS/HOPNC catalyst. (**F**,**G**) AC HAADF-STEM micrographs of the Co-SAS/HOPNC catalyst. Yellow circles represent the isolated single Co atoms. Reproduced from [119], with permission from PNAS, 2018.

**Figure 3 nanomaterials-11-02029-f003:**
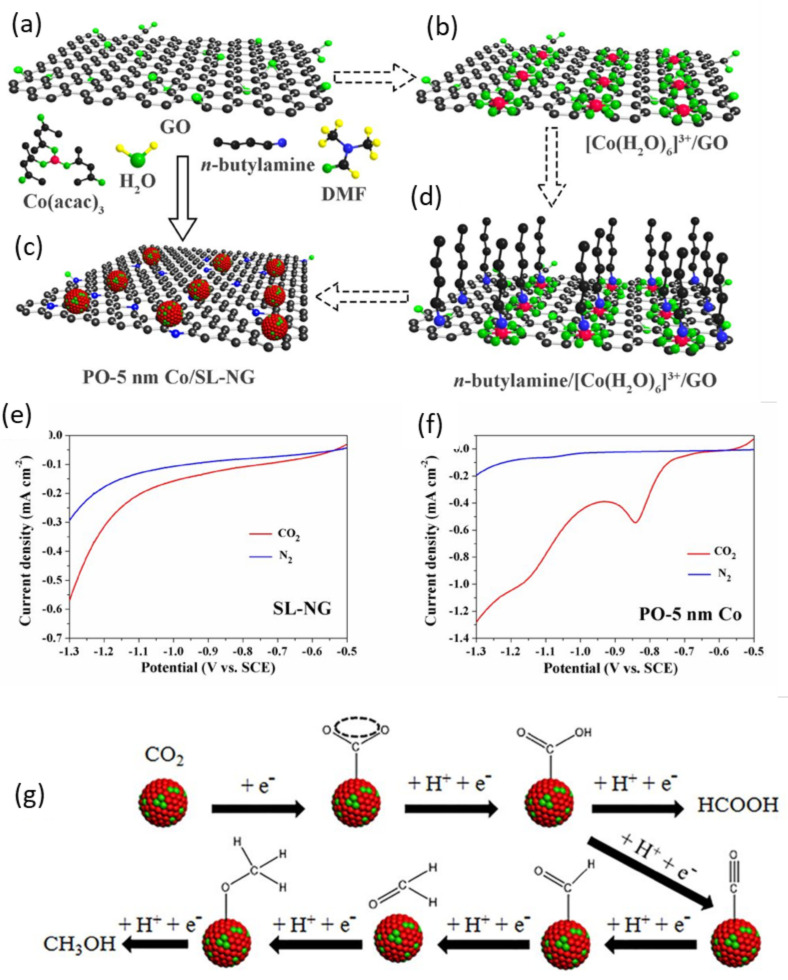
Scheme for the single-step synthesis of PO-5 nm Co/SL-NG (**a**–**d**). Linear-sweep voltammograms of SL-NG catalyst, (**e**) PO-5 nm Co catalyst and (**f**) 30.87 wt% PO-5 nm Co/SL-NG catalyst in saturated CO_2_ (red-line) and saturated-N_2_ (blue-line) 0.1 mol dm^−^^3^ NaHCO_3_ electrolyte and the mechanism of CH_3_OH production (**g**). Reproduced from [120], with permission from American Chemical Society, 2018.

**Figure 4 nanomaterials-11-02029-f004:**
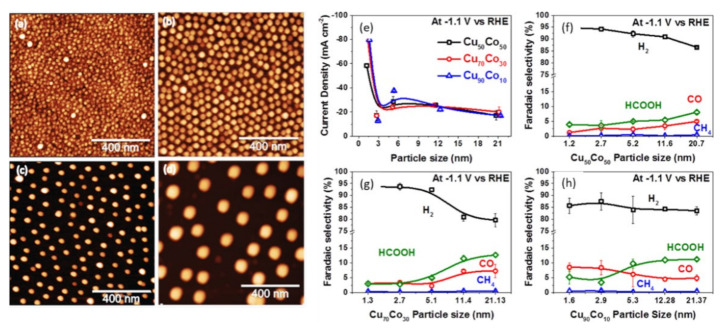
AFM micrographs of Cu50Co50 NPs with different sizes supported on SiO_2_/Si (111). The average NP size: (**a**) 2.7 nm (S2), (**b**) 5.2 nm (S3), (**c**) 11.6 nm (S4), (**d**) 20.7 nm (S5). (**e**) Current density as a function of the particles size of CuxCo100-x NPs. Faradaic selectivity as a function of the NP size: (**f**) bimetallic Cu50Co50 complex, (**g**) Cu70Co30 complex, (**h**) Cu90Co10 complex. The data were obtained after 1 h of CO_2_RR at E  =  −1.1 V vs. RHE. Reproduced from [130], with permission from Elsevier, 2018.

**Figure 5 nanomaterials-11-02029-f005:**
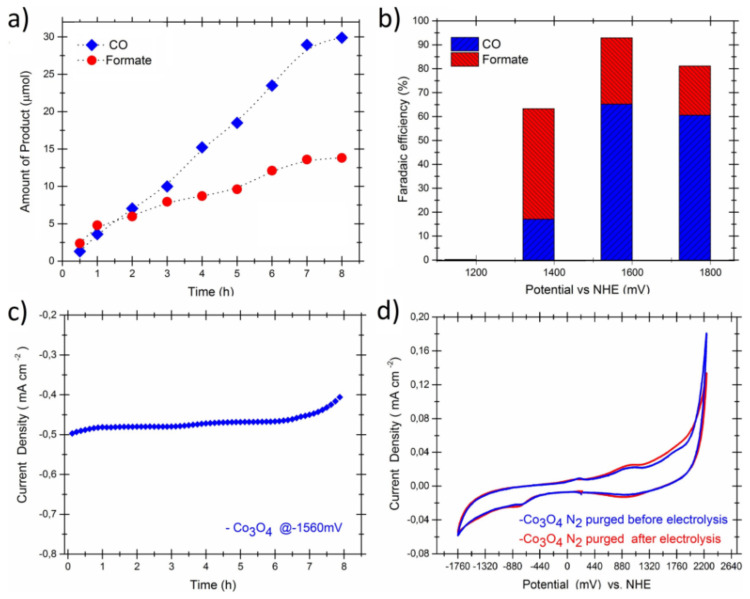
Electroreduction of CO_2_ via the nanofibrous Co_3_O_4_, (**a**) increase in the amount of CO gas and formate as a function of time at a regular electrolysis potential of −1560 mV vs. NHE, (**b**) the electrolysis voltage versus the faradaic efficiency, (**c**) the chronoamperometry results, (**d**) Cyclic voltammograms of the electrode nanofibers measured before and after the electrolysis at a scan rate of 30 mV s^−1^. Reproduced from [131], with permission from Elsevier, 2018.

**Figure 6 nanomaterials-11-02029-f006:**
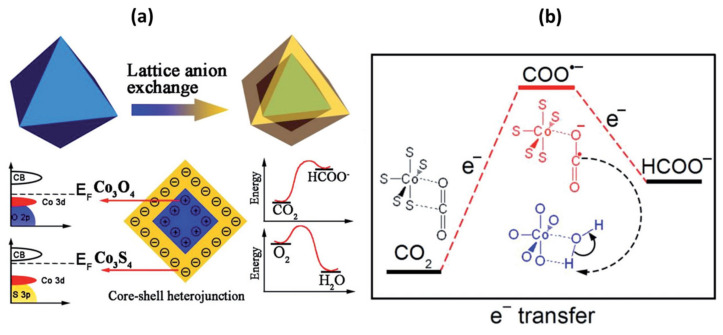
Scheme for the Co_3_S_4_@Co_3_O_4_ core–shell electrocatalyst (**a**) and the mechanistic energy diagram for the formation of HCOO^-^ (**b**). Reproduced from [139], with permission from WILEY-VCH, 2017.

**Figure 7 nanomaterials-11-02029-f007:**
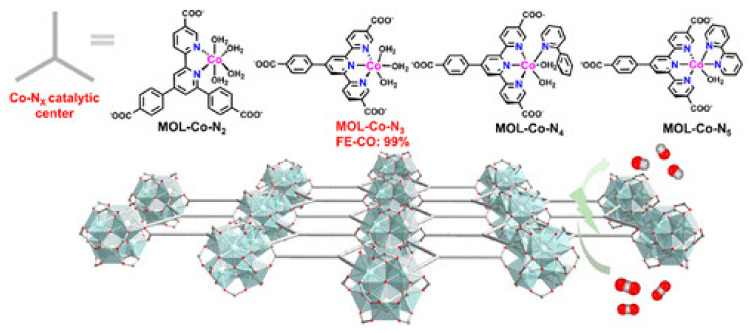
Schematic representation of structure of Co-N_x_(x = 2 − 5) centers in the MOLs. Reproduced from [48], with permission from American Chemical Society, 2016.

**Figure 8 nanomaterials-11-02029-f008:**
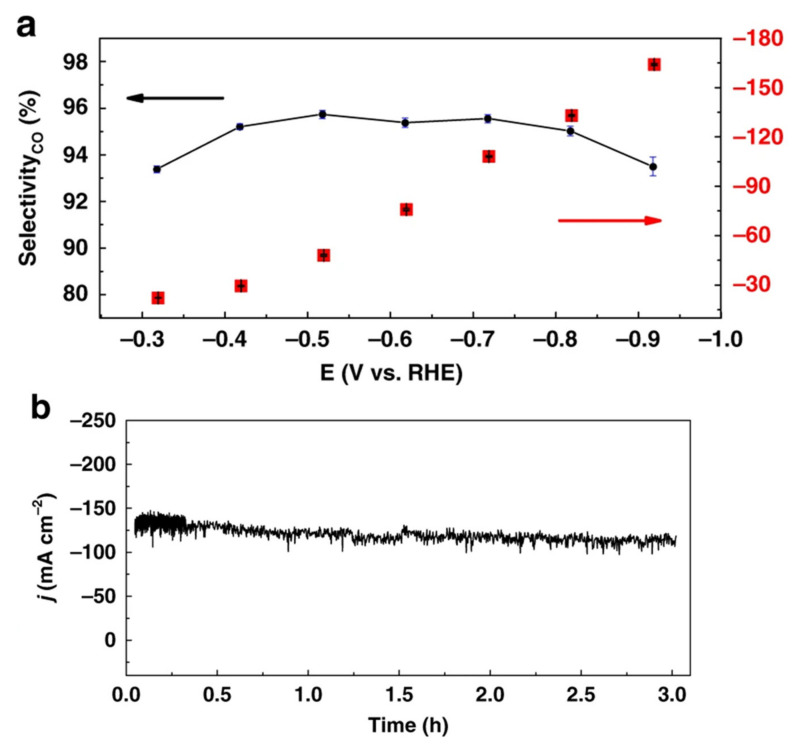
Controlled potential electrolysis and CoPc2 film characterization. (**a**) Current density and selectivity for CO production as a function of the potential and (**b**) bulk electrolysis at a fixed potential (*E*  *=*  −0.72 V vs. RHE) for CoPc2@carbon black deposited onto a carbon paper as cathodic material, in 1 M KOH. c Co K-edge XANES profiles of CoPc2 (black dots) and CoPc2@carbon black before (blue) and after electrolysis (*E*  *=*  −0.72 V vs. RHE) (red) in 1 M KOH solution. Reproduced from [65], with permission from Nature Publishing Group, 2019.

**Figure 9 nanomaterials-11-02029-f009:**
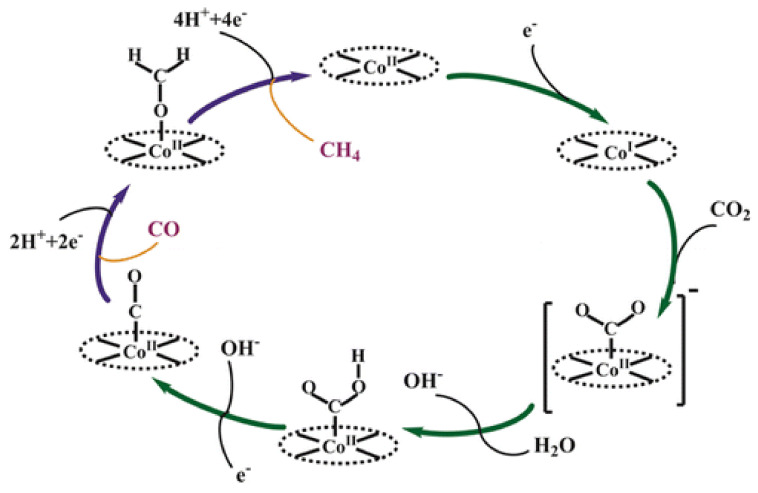
Mechanism of the electrochemical CO_2_ reduction catalyzed via the Co-Porphyrins. Reproduced from [154], with permission from Chemical Society, 2016.

**Figure 10 nanomaterials-11-02029-f010:**
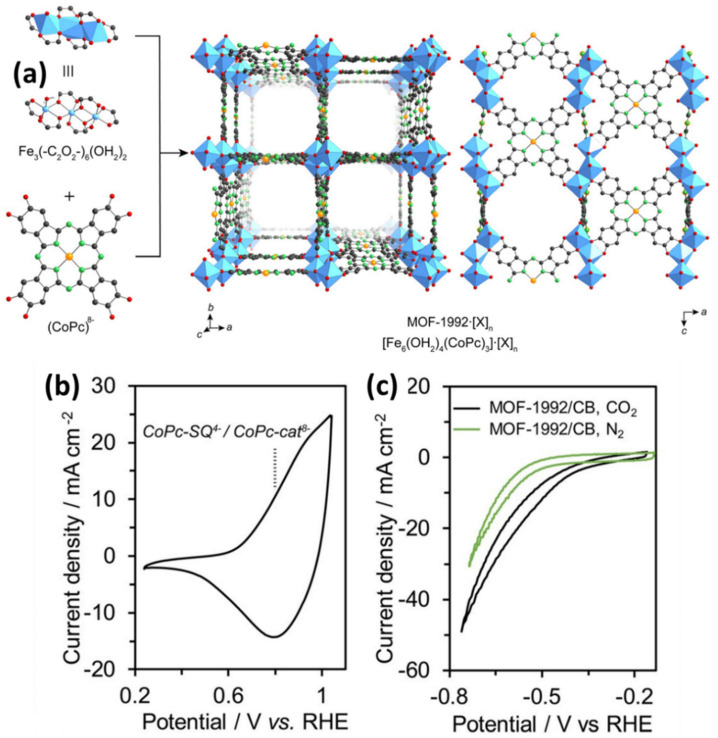
(**a**) Single-crystal X-ray structure of MOF-1992 based on the Fe-trimers and Co-phthalocyanine catechollinkers (CoPc). Atoms color: C-black; O-red; N-green; Co-orange; Fe-blue polyhedra. Hydrogen atoms and the chlorido ligands are omitted. The anionic charge of [Fe_6_(OH_2_)_4_(CoPc)_3_]_6_-, MOF-1992, was balanced by the presence of [X] n counterions (X represents Mg^2+^ or Fe^3+^). Electrochemical characterizations of the MOF-1992 (**b**) Cyclic voltammetry (CV) of the MOF-1992/CB (CB, carbon black). The vertical line display the potential of Co-Pc-semiquinolate (CoPc-SQ) 4-/CoPc-catecholate (CoPc-cat) 8-redox couple (**c**) CV of the MOF-1992/CB in a CO_2_-saturated (black, pH = 6.8) and N_2_-saturated (green, pH = 7.2) KHCO_3_ solution. Reproduced from [165], with permission from American Chemical Society, 2019.

**Table 1 nanomaterials-11-02029-t001:** Performance of cobalt-based electrocatalysts for Electrochemical CO_2_ reduction.

Cobalt Group	Electrocatalysts	Electrolytic Solution	Potential	Current Density (mA/cm^−^^2^)	Main Product	Faradic Efficiency (%)	Stability (h)	Ref
Single atom	Atomic cobalt layers	0.1 M Na_2_SO_4_	−0.85 V vs. SCE	10	COOH	90	60	[167]
	single-atom Co–Tpy-c	0.5 m NaClO_4_	−0.7 to −1.0 V (vs RHE)	46.6	CO	95	24	[116]
	Co/SL-NG	0.1 M NaHCO_3_	−0.9 V vs. SCE	4	CH_3_OH	71.4	10	[120]
	Co_1_-N_4_	0.1 M KHCO_3_	−0.8 V vs. RHE	15.8	CO	82	10	[121]
	Single atom Co-N_5_	0.2 M NaHCO_3_	−0.73 V vs. RHE	6.2	CO	90	10	[43]
Multi metals	Ag-Co	0.5 M KHCO_3_	−2 V vs. SHE	NA	CH_4_	~20	NA	[130]
	Cu-Co	0.1 M KHCO_3_	−1.1 V vs. RHE	30	H_2_	85	NA	[128]
	Co_3_O_4_/Sn	0.5 M KHCO_3_	2 V	NA	HCOOH	92.6	NA	[136]
	Co_3_S_4_ @Co_3_O_4_	0.1 M Na_2_SO_4_	−0.64 V vs. RHE	10	HCOOH	85.3	NA	[139]
	Cu-Co	0.1 M KHCO_3_	−1.1 V vs. RHE	30	H_2_	85	NA	[128]
	Co_3_O_4_/Sn	0.5 M KHCO_3_	2 V		HCOOH	92.6	NA	[136]
	Co_3_O_4_ single-unit-cell layer	0.1 M KHCO_3_	−0.87 V vs. SCE	2.7	COOH	>85	40	[168]
	Ultrathin Co_3_O_4_	0.1 M KHCO_3_	−0.88 V vs. SCE	0.68	COOH	60	20	[132]
	Co_3_O_4_ nanofibers	0.1 M TBAPF_6_ in ACN + 1%_vol_ H_2_O	−1.5 V vs. NHE	10	CO	65	8	[131]
	Co_3_O_4_	0.1 M KHCO_3_	−0.88 V vs. SCE	0.68	HCOOH	60	20	[132]
	CoO anode	0.5 M KHCO_3_	2 V	4.4	C_2_H_5_OH	96.5	NA	[133]
	FeOx/Cox	0.5 M KHCO_3_	−0.55 V vs. RHE	14.49	CO	80	10	[134]
Cobalt nitrides	MOL-Co-N3	1.0 M KHCO_3_	−0.5 V vs. RHE	18.8	CO	99	10	[43]
	700-Co_5.47_N/C	0.5 M NaHCO_3_	−0.7 V vs. RHE	9.78	CO	NA	10	[44]
	CoNC	0.1 M KHCO_3_	−0.6 V versus RHE	4.5	CO	45%	NA	[45]
	Single site catalyst STPyP-Co	0.5 M KHCO_3_	−0.62 V vs. RHE	6.5	CO	96	48	[46]
	Atomically dispersed (Co-N_2_)	0.5 M KHCO_3_	−0.63 V vs. RHE	18.1	CO	94	60	[47]
	700-Co_5.47_N/C	0.5 M NaHCO_3_	−0.7 V vs. RHE	9.78	CO	NA	10	[44]
Cobalt-complexes	Co Complexes	0.1 M NBu4BF4 in DMF + H_2_O	−2 V vs. Fc + /0		COOH	90	1	[64]
	Cobalt phthalocyanine @MWCNT	1 M KOH	−0.92 V vs. RHE	165	CO	99	3	[65]
	Cobalt Phthalocyanine	0.1 M KHCO_3_	−0.8 vs. RHE	3	CO	99	1.3	[67]
	CoPc-P4VP	0.5 M KHCO_3_	−0.68 V vs. RHE	30	CO	>90	12	[68]
	Co(ii) bipyricorrole	DMF 5 M H_2_O	0.35 V vs. Fc + /		CO	75	1	[69]
	Cobalt protoporphyrin	0.1 M HClO4	−0.6 V vs. RHE	0.44	CO	60	1	[70]
	Cobalt protoporphyrin	0.1 M perchlorate	−0.6 V vs. RHE		CO	60	1.3	[70]
	Co-tetraphenylporphyrin-NH_2_	0.1 M NBu4PF6	−2.42 V vs. Ag/AgCl		H2	99	4	[71]
Cobalt MOF/COFs	Metallophthalocyanine (MOF)	KHCO_3_	−0.74 V vs. RHE	17.3	CO	85	10	[121]
	Cobalt porphyrin (COFs)	0.5 M KHCO_3_	–0.67 V vs. RHE	5	CO	90	24	[161]
	Cobalt-porphyrin	0.5 M KHCO_3_	−0.67 V vs. RHE	5.5	CO	76	7	[164]
	MOF-1992	0.1 M KHCO_3_	−0.52 V vs. RHE	16.5	CO	80	6	[165]
	Ag/Co- MOF	0.1 M KHCO_3_	−1.8 V vs. SCE	20	CO	45	10	[166]

## Data Availability

Not applicable.

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
