# Peer review of "Electrochemical Reduction of CO2: A Review of Cobalt Based Catalysts for Carbon Dioxide Conversion to Fuels"

_nanomaterials, 2021, doi:10.3390/nano11082029_

Round 1

Reviewer 1 Report

The authors presented a review of Co based catalysts for carbon dioxide conversion to fuels. This review discussed the recent performance of various Co electrocatalysts, including single-atom Co, Mixed-metal Co,  Co oxide, Co nitrides, Co complex, Co porphyrin, and Co based MOFs and COFs. However, the classification of Co-based catalysts in this manuscript was extremely confusing. Thus, the reviewer could not to recommend this paper for publishing in this journal. The detailed comments to this manuscript are as following, which should be helpful for improving the quality of this work.

  1. The classification of Co-based catalysts in this manuscript was extremely confusing. What does the Co multi-atoms mean and there are no the relevant discussion through the full manuscript. What was the difference among the mixed-metal Co catalysts, Co complex, Co oxide, Co nitrides and Co based MOFs or COFs?
  2. More discussions of Figure 2 should be added in the main manuscript.
  3. The mechanism of ECO2RR for all Co based catalysts should be given and discussed through the manuscript.
  4. The recommendations of 1-4 have already been studied and published in recent years (Lines 660-669, Page 20), therefore, the authors should provide more suggestive recommendation in this part.
  5. The format for the References must be done strictly according the "Guidelines for Authors" (Ref. 1; Ref. 5; etc).
  6. The English of this manuscript needs to be greatly improved.
  • Line 29, Page 1, Cobalt should be changed to cobalt, and the full spelling of MOF and COF should be used. The same problem should be checked carefully through the full manuscript (Line 45, Page 1; Line 87, Page 2;95, Page 2; Line 112, Page 3; Line 143, Page 5, etc).
  • Line 83, Page 2; Line 84, Page 2, CO2R should be changed to CO2RR.
  • The abbreviation of ECO2RR should be firstly labeled at "Electrocatalytic CO2 reduction reaction (ECO2RR)" in Line 73, Page 2.
  • Once the abbreviations of MOFs, COFs, and CO2RR were shown in the previous sentence. Then the abbreviations of MOF and COF must be used (lines 97-98, page 2; Line 104, Page 3; Line 109, Page 3). Please use the abbreviations correctly in this manuscript.
  • The sentence of "Aljabour et al., [101] studied" should be changed to " Aljabour et al. [101] studied" (Line 139, Page 4; Line 131, Page 4; Line 310, Page 10; Line 421, Page 12; Line 459, Page 13; Line 466, Page 13; etc). Please check the full manuscript for this mistake.
  • The abbreviation of PAN should be deleted because that polyacrylnitrile was appeared only once.

Author Response

Reviewer 1

Article: nanomaterials-1308587

Overall review comments

The authors presented a review of Co based catalysts for carbon dioxide conversion to fuels. This review discussed the recent performance of various Co electrocatalysts, including single-atom Co, Mixed-metal Co,  Co oxide, Co nitrides, Co complex, Co porphyrin, and Co based MOFs and COFs. However, the classification of Co-based catalysts in this manuscript was extremely confusing. Thus, the reviewer could not to recommend this paper for publishing in this journal. The detailed comments to this manuscript are as following, which should be helpful for improving the quality of this work.

Response: We are thankful to the reviewer for comprehensively reviewing our paper and provided useful comments for improvement.

Principal comments

  1. The classification of Co-based catalysts in this manuscript was extremely confusing. What does the Co multi-atoms mean and there are no the relevant discussion through the full manuscript. What was the difference among the mixed-metal Co catalysts, Co complex, Co oxide, Co nitrides and Co based MOFs or COFs?

Response:

Thanks for the valuable question. In order to clarify the classification of Co-based
catalysts, we have made changes in the revised manuscript as marked in red. Actually,
the single-atom Co catalysts contain only one Co atom, while multi-atom has been replaced with multi-metals Co catalysts. For clarity, the term “multi-atoms” has been
deleted from abstract. Multi-metal Co catalysts include those catalysts having a single
or multi atoms of Co linked to other metals like Co-Cu, Co-Fe, Co-Ag. Co complex comprises those complexes where Co linked with ligand and act as a catalysts. Co oxide stands for Co3O4. Co nitrides include Co-Nx moieties. While Co-based MOFs
and COFs include those MOFs and Metallo-porphyrin based COFs containing Co atoms. Each of these cobalt compound has certain characteristics and features. We have summerized these Co-based compounds for electrochemical CO2RR.

The classification of Co-based catalyst have been amended in the revised manuscript as marked in red and it is now clear to readers.

  1. More discussions of Figure 2 should be added in the main manuscript.

Response: We are thankful to the reviewer for his comment. More discussion has been added on Figure 2 in the revised manuscript highlighted with red text.

  1. The mechanism of ECO2RR for all Co based catalysts should be given and discussed through the manuscript.

Response: We are thankful to the reviewer for his comment. Several state of the art articles decribed the CO2RR mechanism  [1-4]. This is a mini review which highlight the important application of cobalt based catalysts for CO2RR and the mechanistic study is out of the scope of this review.  However, keeping in mind reviewer comments, we have added mechanism in appropriate positions. As Cobalt based catalysts have mainly selective towards CO, CH4, HCOOH, and CH3OH. Therefore, mechanisms in section 2.1 (Mechanism of CH3OH formation), section 2.4 (Mechanism of COOH formation ), section 2.6 (Mechanism of CO and CH4 formation) are given in the the revised manuscript highlighted with red color.

  1. Lu, Q.; Jiao, F. Electrochemical CO2 reduction: Electrocatalyst, reaction mechanism, and process engineering. Nano Energy 2016, 29, 439-456, doi:https://doi.org/10.1016/j.nanoen.2016.04.009.
  2. Shen, J.; Kolb, M.J.; Göttle, A.J.; Koper, M.T.M. DFT Study on the Mechanism of the Electrochemical Reduction of CO2 Catalyzed by Cobalt Porphyrins. J. Phys. Chem. C 2016, 120, 15714-15721, doi:10.1021/acs.jpcc.5b10763.
  3. Leung, K.; Nielsen, I.M.B.; Sai, N.; Medforth, C.; Shelnutt, J.A. Cobalt−Porphyrin Catalyzed Electrochemical Reduction of Carbon Dioxide in Water. 2. Mechanism from First Principles. The Journal of Physical Chemistry A 2010, 114, 10174-10184, doi:10.1021/jp1012335.
  4. Wang, L.; Chen, W.; Zhang, D.; Du, Y.; Amal, R.; Qiao, S.; Wu, J.; Yin, Z. Surface strategies for catalytic CO2 reduction: from two-dimensional materials to nanoclusters to single atoms. Chem. Soc. Rev 2019, 48, 5310-5349, doi:10.1039/c9cs00163h.

  1. The recommendations of 1-4 have already been studied and published in recent years (Lines 660-669, Page 20), therefore, the authors should provide more suggestive recommendation in this part.

Response: We are thankful to the reviewer for his comment. This is true that the some of the improvement has been done recently. However, still the high current density is needed for higher hydrocarbon production via ECO2RR. For example, the number of electrons involved in the CO2RR are 2,8,12,14 for the formation of CO, methane, ethylene, and ethane, respectively. Keeping in mind we have revised the recommendation part in the revised manuscript.

  1. The format for the References must be done strictly according the "Guidelines for Authors" (Ref. 1; Ref. 5; etc).

Response: We are thankful to the reviewer for his comment. We have checked every reference, which is according to the guidleine of the journal.

  1. The English of this manuscript needs to be greatly improved.

Response: We are thankful to the reviewer for his comment. The English of the manuscrip has been revised again with a native English speaker. All the necessary changes have been made and the English is now improved where indicated and are corrected appropriately.

  1. Line 29, Page 1, Cobalt should be changed to cobalt, and the full spelling of MOF and COF should be used. The same problem should be checked carefully through the full manuscript (Line 45, Page 1; Line 87, Page 2;95, Page 2; Line 112, Page 3; Line 143, Page 5, etc).

Response: We are thankful to the reviewer for his comment. Inside the paragrahs the Cobalt has been changed to cobalt. MOFs and COFs are well-know porous materials. We have added the full name in the start of the review and later used the abbreviations.

  1. Line 83, Page 2; Line 84, Page 2, CO2R should be changed to CO2RR.

Response: We are thankful to the reviewer for pointing this out. We have done this correction in the revised manuscript.

  1. The abbreviation of ECO2RR should be firstly labeled at "Electrocatalytic CO2 reduction reaction (ECO2RR)" in Line 73, Page 2.

Response: We are thankful to the reviewer for his comment. We have used CO2RR in the whole manuscript.

  1. Once the abbreviations of MOFs, COFs, and CO2RR were shown in the previous sentence. Then the abbreviations of MOF and COF must be used (lines 97-98, page 2; Line 104, Page 3; Line 109, Page 3). Please use the abbreviations correctly in this manuscript.

Response: We are thankful to the reviewer for his comment. We have corrected this issue in the whole manuscripts. The full name with abbreviation for MOFs, COFs, and CO2RR were shown in the 1st appearance in the introduction part, and then use abbreviation in later part of the revised manuscript.

  1. The sentence of "Aljabour et al., [101] studied" should be changed to " Aljabour et al. [101] studied" (Line 139, Page 4; Line 131, Page 4; Line 310, Page 10; Line 421, Page 12; Line 459, Page 13; Line 466, Page 13; etc). Please check the full manuscript for this mistake.

Response: We are thankful to the reviewer for his comment. We have checked the whole manuscipt and replaced et al., with et al.

  1. The abbreviation of PAN should be deleted because that polyacrylnitrile was appeared only once.

Response: We are thankful to the reviewer for his comment. One time used used PAN has been deleted in the revised manuscript.

Reviewer 2 Report

In this mini-review, the authors discuss the recent performance of various Cobalt (Co) electrocatalysts, including Co-single atom, Co-multi atoms, Co-complexes, Co-MOFs, Co-COFs, Co-nitrides and Co-oxides in CO2RR. I suggest the review for acceptance after minor revisions listed below.

  1. 1. In the Figure 1, the authors provide the Cobalt as an electrocatalyst in the literature. In addition, the amounts of literatures of Cobalt in CO2RR should be provided.
  2. As for the Co SAC catalyst, the pyrolysis of ZIFs should be emphasized. (ACS Catal. 2018, 8, 3116−3122; Small 2020, 2001896)
  3. More reports about Co particle should be added. For example, Angew. Chem. Int. Ed, 2020, 59, 4914-4919; ACS Appl. Mater. Interfaces 2020, 12, 9307−9315; Journal of CO2Utilization, 2019, 32, 241−250; Catal. Sci. Technol., 2020, 10, 967−977.

Author Response

Reviewer 2

Article: nanomaterials-1308587

Overall review comments

In this mini-review, the authors discuss the recent performance of various Cobalt (Co) electrocaatalysts, including Co-single atom, Co-multi atoms, Co-complexes, Co-MOFs, Co-COFs, Co-nitrides and Co-oxides in CO2RR. I suggest the review for acceptance after minor revisions listed below.

Response: We are thankful to the reviewer for comprehensively reviewing our paper and provided useful comments for improvement.

 Principal comments

  1. 1. In the Figure 1, the authors provide the Cobalt as an electrocatalyst in the literature. In addition, the amounts of literatures of Cobalt in CO2RR should be provided.

Response: We are thankful to the reviewer for his comment. Based on reviewer 2 and reviewer 3 suggestion, we have updated Figure 1 with CO2 reduction literature. The udated figure provid a more meaningful explanation of the coblat based catalysts and cobalt based catalysts for CO2 reduction.

Updated Figure 1. Cobalt as an electrocatalyst and cobalt for CO2 conversion in the literature, source SciFinder, keyword: cobalt electrocatalyst, cobalt catalyst for CO2 reduction

  1. As for the Co SAC catalyst, the pyrolysis of ZIFs should be emphasized. (ACS Catal. 2018, 8, 3116−3122; Small 2020, 2001896)

Response: We are thankful to the reviewer for recommending important publications. ACS Catal. 2018, 8, 3116−3122 [1]; and Small 2020, 2001896 [2] articles has been added in the revised manuscripts.

  1. More reports about Co particle should be added. For example, Angew. Chem. Int. Ed, 2020, 59, 4914-4919 [3]; ACS Appl. Mater. Interfaces 2020, 12, 9307−9315 [4]; Journal of CO2Utilization, 2019, 32, 241−250 [5]; Catal. Sci. Technol., 2020, 10, 967−977 [6].

Response: We are thankful to the reviewer for the important recommendataion. All the suggested articles has been added in the revised manuscript.

  1. Pan, F.P.; Zhang, H.G.; Liu, K.X.; Cullen, D.; More, K.; Wang, M.Y.; Feng, Z.X.; Wang, G.F.; Wu, G.; Li, Y. Unveiling Active Sites of CO2 Reduction on Nitrogen-Coordinated and Atomically Dispersed Iron and Cobalt Catalysts. ACS Catal 2018, 8, 3116-3122, doi:10.1021/acscatal.8b00398.
  2. Hou, P.; Song, W.; Wang, X.; Hu, Z.; Kang, P. Well-Defined Single-Atom Cobalt Catalyst for Electrocatalytic Flue Gas CO2 Reduction. Small 2020, 16, 2001896, doi:https://doi.org/10.1002/smll.202001896.
  3. He, C.; Zhang, Y.; Zhang, Y.; Zhao, L.; Yuan, L.P.; Zhang, J.; Ma, J.; Hu, J.S. Molecular evidence for metallic cobalt boosting CO2 electroreduction on pyridinic nitrogen. Angew. Chem. 2020, 132, 4944-4949, doi:https://doi.org/10.1002/ange.201916520.
  4. Daiyan, R.; Chen, R.; Kumar, P.; Bedford, N.M.; Qu, J.; Cairney, J.M.; Lu, X.; Amal, R. Tunable Syngas Production through CO2 Electroreduction on Cobalt–Carbon Composite Electrocatalyst. ACS Appl. Mater. Interfaces 2020, 12, 9307-9315, doi:10.1021/acsami.9b21216.
  5. Liu, W.; Miao, Z.; Li, Z.; Wu, X.; Zhou, P.; Zhao, J.; Zhao, H.; Si, W.; Zhou, J.; Zhuo, S. Electroreduction of CO2 catalyzed by Co@N-C materials. J. CO2 Util 2019, 32, 241-250, doi:https://doi.org/10.1016/j.jcou.2019.04.005.
  6. Miao, Z.; Liu, W.; Zhao, Y.; Wang, F.; Meng, J.; Liang, M.; Wu, X.; Zhao, J.; Zhuo, S.; Zhou, J. Zn-Modified Co@N–C composites with adjusted Co particle size as catalysts for the efficient electroreduction of CO2. Catalysis Science & Technology 2020, 10, 967-977, doi:10.1039/c9cy02203a.

Reviewer 3 Report

The authors summarize the advancement of Co-based catalysts for carbon dioxides reduction catalysis, including Co single atoms, Co cluster, Co complexes, Co-MOF, Co-COFs, Co nitrides and Co-oxides. Finally, the authors give a perspective for the development of efficient CO2 reduction reactions. I would like to recommend minor revision before acceptance. My specific comments are as follows.

  1. The content of Figure 1 is not directly related to CO2 reduction. Please use the other figures to replace this one.
  2. The authors summarized many Co-based compounds for CO2 reduction. However, some important Co-based materials are missed, such as cobalt sulfides, selenides. please add a paragraph to discuss other Co-based materials.
  3. In the introduction part, the authors should point out the possible advantages using Co over other metal-based materials. 

Author Response

Reviewer 3

Article: nanomaterials-1308587

Overall review comments

The authors summarize the advancement of Co-based catalysts for carbon dioxides reduction catalysis, including Co single atoms, Co cluster, Co complexes, Co-MOF, Co-COFs, Co nitrides and Co-oxides. Finally, the authors give a perspective for the development of efficient CO2 reduction reactions. I would like to recommend minor revision before acceptance. My specific comments are as follows.

Response: We are thankful to the reviewer for comprehensively reviewing our paper and provided useful comments for improvement.

Principal comments

  1. The content of Figure 1 is not directly related to CO2 reduction. Please use the other figures to replace this one.

Response: We are thankful to the reviewer for his comment. Based on reviewer 2 and reviewer 3 suggestion, we have updated Figure 1 with CO2 reduction literature. The udated figure provid a more meaningful explanation of the coblat based catalysts and cobalt based catalysts for CO2 reduction.

Updated Figure 1. Cobalt as an electrocatalyst and cobalt for CO2 conversion in the literature, source SciFinder, keyword: cobalt electrocatalyst, cobalt catalyst for CO2 reduction

  1. The authors summarized many Co-based compounds for CO2 reduction. However, some important Co-based materials are missed, such as cobalt sulfides, selenides. please add a paragraph to discuss other Co-based materials.

Response: We are thankful to the reviewer for his comment. Co-based sulfides and selenides are widely used for HER, OER and Photocatalysis and dyesensitization. However, these compounds are not employed in ECO2RR. Since, the literature for electrocatalytic CO2 reduction over the Co-based sulfide and selenide compounds is limited that’s why we didn’t mentioned these materials in our review article.

  1. In the introduction part, the authors should point out the possible advantages using Co over other metal-based materials. 

Response: We are thankful to the reviewer the valuable suggestion. The advantages of Co-based catalysts over other metal-based materials have provided in the revised manuscript as marked in red.

Co-based materials have many advantages over others because as a popular metal, Co belongs to the group VIII B of periodic table. Co as an earth-abundant transition metal is a splendid alternative to the noble metals such as Pt, Ir, Ru etc. For CO2 reduction, Co has been used as a prominent source for fabricating noble metal free electro/photocatalysts due to fascinating properties such as loosely bonded d-electrons and therefore easily available multiple oxidation states (Co(0), Co(I), Co(II), Co(III) and Co(IV). Moreover, it is found that a transition from Co(II) to Co(I) is involved at intermediate state for CO2 reduction. Hence, high activity, outstanding stability and product selectivity is achieved through Co based catalysts for CO2 reduction [1, 2].  Cobalt is found to be more reactive as compared to other earth-abundant metals such as Fe, Ni, Cu etc. due to the possession of modest CO2 adsorption and d-band closeness to the Fermi level [3].  Co based catalysts have been explored as effective cathode materials for electroreduction of CO2 to CO exhibiting high activities and selectivity [4]. For CO2 photoreduction, the presence of Co metal sites in Co based MOFs offer the traps for electrons for facilitation in electrons-holes separation thus, providing a longer lifetime for electrons for reduction reaction. Co is found to be an important stabilizer for major intermediates in CO2 reduction [2, 3]. Co-based materials have applications in various other fields such as energy storage, catalysis, and thermopower. Co-based materials (i.e., NaxCoO2) play a critical role in cathode and anode materials for Na-ion batteries. Likewise, LiCoO2 has been regarded as one of the most commercialized cathode materials for Li-ion batteries. With respect to anode materials, cobalt oxides and cobalt chalcogenides exhibit high theoretical capacity for sodium storage [5]

  1. Yang, P., et al., Cobalt Nitride Anchored on Nitrogen-Rich Carbons for Efficient Carbon Dioxide Reduction with Visible Light. 2021. 280: p. 119454.
  2. Li, C., et al., Carbon dioxide photo/electroreduction with cobalt. Journal of Materials Chemistry A, 2019. 7(28): p. 16622-16642.
  3. Nguyen, D.-T., C.-C. Nguyen, and T.-O. Do, Rational one-step synthesis of cobalt clusters embedded-graphitic carbon nitrides for the efficient photocatalytic CO2 reduction under ambient conditions. Journal of Catalysis, 2020. 392: p. 88-96.
  4. Zhu, M., et al., Inductive and electrostatic effects on cobalt porphyrins for heterogeneous electrocatalytic carbon dioxide reduction. Catalysis Science & Technology, 2019. 9(4): p. 974-980.
  5. Qi, S., et al., Cobalt-based electrode materials for sodium-ion batteries. Chemical Engineering Journal, 2019. 370: p. 185-207.

Reviewer 4 Report

The problems of air pollution can be partially solved by progress in catalysis techniques and materials.
Authors represented a very good structured review of Cu containing catalysts.
Since such papers are based on previous works, the citations should be done more carefully.
Please, add corresponding citations for every work that you mentioned, some examples are:
".....ith different Cu loading (i.e. x=50, 70, 90 for CuxCox-100 system) by Bernal et al., 2018."
"Hwang and coworkers found that turning the perovskite chemistry will result...."
".... the neighborhood of e-MPc.Guo et al., 2017 introduced the concept..."
"...Gao et al., 2017 investigated the..."
Please, support this statement by additional reference "Various type of catalysts have been reported for the conversion of waste into useful products including zeolites [12-25], metal and metal oxides [26-35, DOI: 10.3390/NANO10112183],.. "  
Please, add definition of "Faradaic  efficiency"  in the text and if you decide to write the first letter capital, use this style through the text (line 505).
Line 542: you have double "for" in the section name. 

This statement of lines 320-324 from 2.5. belongs to Introduction section

Please, add a copyright information to figure 9.

This review is comprehensive and interesting. It could be published after minor revisions.

Author Response

Reviewer 4

Article: nanomaterials-1308587

Overall review comments

This review is comprehensive and interesting. It could be published after minor revisions.

The problems of air pollution can be partially solved by progress in catalysis techniques and materials.
Authors represented a very good structured review of Cu containing catalysts.

Response: We are thankful to the reviewer for comprehensively reviewing our paper and provided useful comments for improvement.

Principal comments

  1. Since such papers are based on previous works, the citations should be done more carefully.
    Please, add corresponding citations for every work that you mentioned, some examples are:
    ".....ith different Cu loading (i.e. x=50, 70, 90 for CuxCox-100 system) by Bernal et al., 2018."
    "Hwang and coworkers found that turning the perovskite chemistry will result...."
    ".... the neighborhood of e-MPc.Guo et al., 2017 introduced the concept..."
    "...Gao et al., 2017 investigated the..."

Response: We are thankful to the reviewer for his comment.The mentioned articles have been cited in the revised manuscript.

  1. Please, support this statement by additional reference "Various type of catalysts have been reported for the conversion of waste into useful products including zeolites [12-25], metal and metal oxides [26-35, DOI: 10.3390/NANO10112183],.. "  

Response: We are thankful to the reviewer suggestion. The suggested article has been added in the revised manuscript.

  1. Please, add definition of "Faradaic  efficiency"  in the text and if you decide to write the first letter capital, use this style through the text (line 505).

Response: We are thankful to the reviewer for his comment. We have used the 1st letter capital in the whole text of the revised manuscript as “Faradic effeciency”. We believe that everyone working in ECO2RR understnd the definition of Faradic effeceincy, so there is no need to add it. If reviewer, still want to add the difinition, then we shall add in the furutre revision.

  1. Line 542: you have double "for" in the section name. 

Response: We are thankful to the reviewer for his comment. We have deleted the consective second “for” in the revised manuscript.

  1. This statement of lines 320-324 from 2.5. belongs to Introduction section

Response: We are thankful to the reviewer for his comment. These lines have been removed and added into the inroduction section.

  1. Please, add a copyright information to figure 9.

Response: We are thankful to the reviewer for his comment. The menionted Figure copyright has been added in the revised manuscript.

Round 2

Reviewer 1 Report

The authors have addressed all my raised issues, therefore, this manuscript can be published without any further modification.